# SocioSim: A Framework for Rapid, Policy-Relevant Audience Simulation

## Abstract

Recent advancements in Large Language Models (LLMs) have enabled complex social simulations featuring persistent, interactive agents. While these models offer high-fidelity insights into emergent behaviors, they are often computationally intensive and ill-suited for rapid exploration of public attitudes or policy reception. We introduce SocioSim, a complementary framework designed for the rapid simulation of large, demographically-nuanced audiences. Instead of modeling longitudinal agent interactions, SocioSim employs a multi-stage LLM pipeline to generate cross-sectional insights. This paper details the SocioSim methodology, which consists of three core stages: Persona Blueprint Definition, Iterative Persona Instantiation, and Contextualized Response Generation. We present a case study demonstrating how the framework was used to model societal attitudes toward AI companionship, revealing a non-obvious generational divide in public sentiment. SocioSim provides a novel, structured method for researchers and policymakers to generate hypotheses, de-risk research, and anticipate public reaction to emerging social and technological phenomena.

## 1 Introduction

The capacity of Large Language Models (LLMs) to simulate human-like behavior has unlocked new frontiers in computational social science. Frameworks for creating generative agents that inhabit and interact within simulated environments have shown remarkable potential for studying emergent social dynamics, from cultural evolution to complex decision-making processes (Park et al. (2023); Perez et al. (2024)). Projects like Project Sid and Election Sim demonstrate the ambition to model entire societies, providing powerful tools to explore collective behavior at an unprecedented scale (Altera et al. (2024); Zhang et al. (2024)).

However, these high-fidelity, multi-agent interactive simulations, while powerful, present significant operational challenges. They are often computationally expensive and require extended run-times, making them less suitable for tasks requiring rapid feedback on specific, isolated questions. For instance, a policymaker wanting to gauge public reaction to a new technology, or a researcher testing the framing of a sensitive topic, requires a different class of tool—one geared towards speed and targeted insight rather than longitudinal emergent behavior.

This paper introduces SocioSim, a novel framework designed to fill this methodological gap. SocioSim leverages LLMs not to create persistent, interacting agents, but to simulate the collective attitudinal responses of a large, well-defined audience to a specific stimulus. Its novelty lies in a structured, multi-stage pipeline that separates the definition of an audience's structure from the generation of its individual members, ensuring both consistency and diversity. It functions as an on-demand, digital focus group, allowing for rapid hypothesis testing.

In this paper, we:

1. Describe the three-stage architecture of the SocioSim framework.
2. Present a case study on AI companionship that showcases the framework's ability to uncover non-obvious social dynamics.

3. Discuss the framework's position within the broader landscape of social simulation, including its ethical considerations and approach to validation.

## 2 The SocioSim Framework

SocioSim's methodology is an orchestrated pipeline that ensures structural integrity and response heterogeneity through distinct, sequential stages. The process is managed via a series of LLM calls, with outputs at each stage validated against dynamic Pydantic models.

### 2.1 Stage 1: Persona Blueprint Definition

The process begins not with generating characters, but with defining their *structure*. A researcher provides a high-level audience description (e.g., "The general public with diverse demographics and tech familiarity"). This description is passed to an LLM (Gemini 2.5 Flash), which is prompted to act as an expert persona designer. The LLM's task is to generate a `CharacterProfileOutput`, a structured JSON object defining the key `profile_fields` (e.g., 'age_group', 'tech_savviness', 'parental_status') necessary to model this audience. For each field, the LLM specifies its type (e.g., `single_choice`), description, and potential options. This `CharacterProfileOutput` serves as a validated, reusable "blueprint" for all subsequent characters.

### 2.2 Stage 2: Iterative Persona Instantiation

With the blueprint established, the framework generates a list of unique character instances. This is an iterative and batched process designed to encourage diversity. The system makes parallelized calls to an LLM (Gemini), with each call prompted to generate a small batch of characters conforming to the Pydantic schema derived from the blueprint. Crucially, the prompt for each new batch includes a sample of *previously generated characters*. This feedback mechanism instructs the LLM to avoid repetition and generate profiles that occupy different points in the defined attribute space. If any generated character fails validation, a simpler LLM is used to attempt a structural fix, maximizing the yield of usable personas. This stage results in a list of N unique, validated character profiles.

### 2.3 Stage 3: Contextualized Response Generation

In the final stage, each character profile is used to simulate a survey response. For each of the N characters, a dedicated LLM instance is prompted with a system message that provides the character's full profile as its persona. The prompt includes the full survey text and instructs the model to answer from the perspective of its assigned persona. The LLM's JSON output is validated against a dynamically created Pydantic `SurveyModel`. This one-to-one mapping of persona-to-respondent ensures that each answer is grounded in a specific, detailed profile, rather than emerging from a generic model. The result is a complete dataset of N individual survey responses ready for analysis.

## 3 Case Study: Societal Apprehension of AI Companionship

To demonstrate the framework's ability to uncover nuanced social dynamics, we ran a simulation exploring public sentiment towards AI companions.

**Objective:** To gauge initial public reaction to AI designed for friendship and support, and identify key demographic drivers of these attitudes.

**Methodology:**

- **Audience:** We defined a general population audience (N=897), prompting the framework to create a blueprint with fields for age group, tech savviness, and parental status.

| Gut Reaction | Boomers (61+) | Gen Z (18-28) |
|---|---|---|
| Very Creepy | **60.1%** | 1.1% |
| Neutral / Unsure | 1.7% | 48.8% |
| Comforting | 0.0% | 14.7% |

Table 1: Percentage of gut reactions within each generation, highlighting the stark divide in sentiment towards AI companions.

- **Instrument:** An AI-assisted survey was designed to probe reactions, perceived benefits, and ethical concerns regarding AI companions.

**Results:** The simulation revealed a powerful, non-obvious generational divide. While a majority (80.9%) of the total population found the concept "Creepy" or "Very Creepy," this sentiment was not evenly distributed. As shown in Table 1, older generations were highly skeptical, while younger generations were significantly more open. Among Boomers (61+), 60.1% of their cohort found the concept "Very Creepy," compared to only 1.1% of Gen Z (18-28). Conversely, of all respondents who found the concept "Comforting," 66.7% were from Gen Z.

This finding is more nuanced than a simple "people are scared of AI" conclusion. It provides a specific, testable hypothesis about a key demographic cleavage that will shape the technology's societal adoption.

## 4 Discussion

### 4.1 On Validation and Fidelity

We position SocioSim as a tool for *hypothesis generation* and *pre-research exploration*, not as a replacement for studies with human participants. The validity of its outputs is contingent on the underlying LLM's capacity to model human attitudes. While this is an active area of research, recent studies provide a strong basis for exploring this methodology. Work by Argyle et al. (2023) and Kim & Lee (2024) has shown remarkable correspondence between LLM-generated responses and real-world survey data, with correlations often exceeding 0.90. Similarly, Aher et al. (2023) demonstrated that LLMs can replicate classic findings from human subject experiments. SocioSim is a framework built to leverage this emerging capability for rapid, low-cost exploration, with the explicit understanding that its findings must be treated as well-informed hypotheses to be validated through traditional empirical methods.

### 4.2 Ethical Considerations and Future Work

The ethical challenges of LLM-based simulation, particularly the risk of misrepresenting or "flattening" identity groups (Wang et al. (2025)), are significant. We do not frame SocioSim as a solution to this problem, but rather as a potential tool for *studying* it. By allowing researchers to define specific audiences and observe how the LLM simulates their attitudes, the framework can be used to probe the model's internal biases and representations of different social groups in a controlled environment (Hu et al. (2025)). This can help surface potential issues in the model itself before they are deployed in more consequential applications.

The primary limitation of SocioSim remains its dependence on the fidelity of the underlying LLM. Future work will involve systematic comparisons of simulation outputs against real-world survey data to calibrate the model and better quantify its domain-specific accuracy.

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
