# OpenReview forum: "SocioSim: A Framework for Rapid, Policy-Relevant Audience  Simulation"
_colmweb.org/COLM/2025/Workshop/Social_Sim — Social Sim'25_

### Official Review · Reviewer_4Mhh · 2025-07-17
**A promising framework for rapid audience simulation that suffers from lack of empirical validation and insufficient evaluation depth.**

**Rating:** 5
**Overall Assessment:** 2
**Confidence:** 4

**Review:**

This paper addresses a need for rapid audience simulation in policy research through a well-structured three-stage pipeline. The approach is novel and differentiates this work from existing multi-agent simulations. The The case study also shows interesting demographic patterns that demonstrate potential utility for hypothesis generation.

However, the work has significant limitations that prevent it from being a strong contribution. While the paper is generally well-written with clear methodology description, it lacks critical implementation details and validation against real human data and other related methods. The distinction between "cross-sectional insights" and multi-agent approaches is never adequately explained beyond computational cost, creating conceptual gaps that limit understanding of the framework's value proposition.

### Pros:
- The paper introduces a novel, structured three-stage pipeline that provides a systematic approach to audience simulation by clearly separating persona design from response generation.
- The framework demonstrates clear practical applications for policy research and hypothesis generation, directly addressing genuine needs within the field of computational social science.
- If the framework is properly validated, it could become a valuable tool for rapid policy assessment.

### Cons:
- The paper completely lacks validation against real human survey data, even though it cites several related works in the area.
- The evaluation is limited to a single case study with a narrow scope, which is insufficient to demonstrate the framework’s robustness across different domains, question types, or demographic compositions.
- Essential implementation details are missing, including the exact prompt templates, model parameters, computational costs, and descriptions of potential failure modes.
- The case study cannot be reproduced because the paper does not provide details on how demographic groups were defined, how personas were distributed across categories, or what the actual survey questions were.
- The authors use terms like "gut reactions" without providing proper definitions, and the survey questions are not included.
- The demographic modeling in the framework is overly simplistic, as it relies on only three fields (age, tech savviness, and parental status) to represent human populations.
- The practice of including previous characters in generation prompts may introduce systematic biases, but the paper does not investigate this possibility.
- There is no dedicated literature review section, even though related work is discussed throughout the paper.
- The statistical analysis presented in the paper does not include confidence intervals, significance tests, or any measures of uncertainty.
- The framework is not described with any formal mathematical notation or equations.
- The paper does not sufficiently explain what is meant by "cross-sectional insights" or how this approach differs methodologically from multi-agent simulations, aside from mentioning computational cost.

**Comments Suggestions And Typos:**

The authors are encouraged to conduct a systematic comparison between SocioSim outputs, real human survey data across, and other simulation frameworks across multiple domains to establish baseline accuracy metrics. This will strengthen the framework's credibility and help identify where the three-stage pipeline might introduce unique biases compared to simpler approaches.

The paper would also benefit from better presentation and conceptual clarity. A visual diagram showing the three-stage pipeline flow would help readers understand the methodology, and a dedicated literature review section would better position this contribution within the current research in social simulation. The authors also need to clearly explain what "cross-sectional insights" means and how this approach differs methodologically from multi-agent simulations beyond just computational cost.

For reproducibility, the authors should provide complete implementation details including the actual survey questions used in the case study, exact demographic group definitions, persona distribution across categories, and the specific generation prompts employed. The demographic modeling should be expanded beyond the current three fields to better represent complex human populations.
Similarly, several technical details need clarification: exact LLM model versions, computational costs, and confidence intervals for Table 1. Some sentences throughout the paper sound artificially generated (e.g. This finding is more nuanced than a simple ”people are scared of AI” conclusion. It provides a specific, testable hypothesis about a key demographic cleavage that will shape the technology’s societal adoption.)
These sentences would benefit from more natural phrasing to improve readability.
A formal mathematical description of the framework would also be valuable.

The authors should also investigate how including previous characters in generation prompts affects diversity and whether this introduces systematic biases. Additionally, providing examples of generated personas would help readers assess the quality and diversity of the framework's output.

**Ethical Concerns:**

SocioSim risks perpetuating and amplifying the biases inherent in LLM training data, potentially misrepresenting or stereotyping diverse social groups through simplified demographic schemas. The use of only three demographic fields to represent complex human populations might also reinforce harmful stereotypes.

Without proper safeguards and validation, the framework could be deployed for manipulation or propaganda purposes or even enable opinion manipulation if the outputs are treated as representative of real human attitudes rather than as preliminary hypotheses requiring validation.

The authors acknowledge some of these concerns but provide limited concrete mitigation strategies or ethical guidelines for responsible deployment.

**Paper Summary:**

The paper introduces SocioSim, a framework that uses LLMs to simulate survey responses from large audiences with specific demographics. The authors claim their approach provides "cross-sectional insights" as an alternative to computationally expensive multi-agent simulations. The framework operates through three stages. Initially, an LLM generates structured demographic profiles. Then, batches of diverse character instances are generated based on the generated profiles in the previous stage. Finally, each persona is used to generate a survey response to a specific prompt.

The paper includes a case study on AI companionship with 897 simulated responses, showing a generational divide: 60.1% of "Boomers" (61+) vs. 1.1% of "Gen Z" (18-28) found AI companions "Very Creepy."
The authors position SocioSim as a tool for hypothesis generation and pre-research exploration rather than a replacement for human studies.

**Relevance:**

5

**Summary Of Strengths:**

The paper presents a novel structured pipeline that provides a systematic approach to audience simulation through its three-stage methodology. The framework has clear practical applications for policy research and hypothesis generation, addressing real needs in computational social science. The methodology is well-presented, and the case study findings reveal interesting demographic patterns that demonstrate the framework's potential utility.

**Summary Of Weaknesses:**

The most significant limitation is the complete absence of validation against real human survey data and other frameworks. While the authors cite previous work showing strong correlations between LLM responses and human surveys, they provide no direct validation of their specific framework's accuracy. SocioSim's pipeline and the addition of the feedback mechanism could introduce unique biases not present in simpler approaches.

The evaluation is insufficient for a framework claiming broad applicability. The single case study on AI companionship, while interesting, cannot demonstrate the system's robustness across different domains, question types, or demographic compositions. The case study itself lacks essential reproducibility details, as it is not mentioned how the demographic groups were defined, how the 897 personas were distributed across categories, or what the actual survey questions were. Terms like "gut reactions" are also used without definition.

The authors also provide no information about prompt engineering strategies, model parameters, computational costs, or failure modes. The demographic modeling appears overly simplistic, as only three fields (age, tech savviness, parental status) are used to represent human populations. The inclusion of previous characters in generation prompts, while intended to encourage diversity, could introduce systematic biases that aren't investigated.

The paper also suffers from conceptual gaps that limit understanding. The distinction between "cross-sectional insights" and multi-agent approaches is never clearly explained beyond computational cost. The statistical analysis lacks confidence intervals or significance tests. While related work is discussed throughout, the absence of a dedicated literature review section makes it difficult to assess how this contribution fits within the broader landscape of social simulation research.

---

### Official Review · Reviewer_CXYG · 2025-07-18
**Review for Submission 7**

**Rating:** 6
**Overall Assessment:** 3
**Confidence:** 4

**Review:**

See below

**Comments Suggestions And Typos:**

Paper is well written

**Paper Summary:**

The authors introduce SocioSim, a three‑stage LLM‑based pipeline that simulates the survey responses of a large, demographically‑defined audience without running costly multi‑agent interactions. Stage 1 lets an LLM create persona blueprints, Stage 2 instantiates hundreds of unique personas to encouraging diversity, and Stage 3 prompts an LLM to answer a survey from each persona’s perspective. The paper presents this framework as being a fast, policy‑oriented alternative to high‑fidelity agent societies and discusses validation and ethical issues .

**Relevance:**

4

**Summary Of Strengths:**

- The framework is interesting, and the three components make sense intuitively.
- The AI companionship case study is interesting and relevant from an HCI research standpoint.

**Summary Of Weaknesses:**

The diversity prompts may output a skewed distribution which could affect the ecological validity of the simulations, but no error analysis or robustness tests are conducted in the paper or reported.
The paper does not validate actually validate its own outputs against any human survey, which makes it hard to understand how well it represents realistic scenarios across domains.

---

### Meta-Review · Area_Chair_4tXB · 2025-07-21

**Recommendation:** Accept

**Metareview:**

Please take a look at the reviews and incorporate their feedback.